# Effect of Irrigation on Crop Yield and Nitrogen Loss in Simulated Sloping Land with Shallow Soils

**DOI:** 10.3390/plants14172666

**Published:** 2025-08-26

**Authors:** Haitao Liu, Chaowen Lin, Li Yao, Hong Wang, Shanghong Chen, Lufang Yang

**Affiliations:** 1Sichuan Academy of Agricultural Sciences Institute of Resources and Environment, Chengdu 610066, China; lcw-11@163.com (C.L.);; 2Sichuan Academy of Agricultural Sciences Ziyang Experimental Station, Ziyang 641300, China

**Keywords:** sloping land, irrigation, crop production, runoff, nitrogen loss

## Abstract

Seasonal drought and nitrogen loss through runoff are two critical problems in the sloping land with shallow soils in southwest China. Irrigation is an effective way to alleviate drought and increase crop yields. Although irrigation is a proven strategy to mitigate drought stress and enhance yields, increased soil moisture under irrigation may exacerbate water and nitrogen losses. Therefore, this study aimed to investigate the long-term effects of irrigation regimes on crop yield, surface runoff, leaching, and nitrogen loss in shallow soil systems. Three experimental treatments were implemented: rainfed control (RF), single irrigation at a flowering stage (SI), and full irrigation (FI). The annual crop yield under SI and FI treatments was 16.4% and 43.5% higher than treatment RF, respectively. The surface runoff in RF was 46.2% and 52.8% higher than the values in SI and FI, respectively. Conversely, the leaching water volume in RF was 13.7% and 13.6% lower than in SI and FI, respectively. The total runoff did not differ significantly, as reduced surface runoff offset elevated leaching. The annual nitrogen loss was 35.4, 30.5, and 22.0 kg N ha^−1^ in RF, SI, and FI treatments, respectively. Irrigation can significantly decrease the nitrogen loss. Leaching accounted for 96% of the total nitrogen loss. Enhanced crop nitrogen uptake under irrigation reduced total nitrogen concentrations in both soil and leaching water solution, which was the main factor for the decrease in total nitrogen loss under irrigation. These results indicate that in sloping land with shallow soil layers, optimal irrigation scheduling can effectively enhance crop yield without elevating nitrogen leaching risks. The study provides a scientific basis for formulating irrigation strategies in the study region.

## 1. Introduction

Due to the rising global population and changes in dietary habits, there has been a significant surge in the demand for food, particularly in developing countries [1,2]. Due to limited farmland resources in the plains for crop production, achieving high and stable yields in sloping land are crucial for world food security [3,4]. In southwest China, sloping land is a dominant agricultural landscape, supporting grain and food for millions of people. Ensuring high and stable yields from these lands is critical not only for local livelihoods but also for the sustainability of entire river basin ecosystems.

The soil of sloping land in southwest China develops from purple sedimentary rock and limestone. Due to the serious soil erosion, the soil layer is very shallow, 75% of which ranges from 15 to 75 cm [5]. The sloping farmland with shallow soils in southwest China faces two challenges: seasonal drought and serious soil water and nutrient losses. The crop production is rainfed in this area. Because of the shallow soil layer and shortage of soil water capacity, droughts occur frequently in dry seasons [6], and the yield is low. Steep slopes exacerbate surface runoff, leading to soil erosion and the loss of essential nutrients [7]. The nitrogen and phosphorus nutrient loss through runoff from the sloping land pollutes the surface water [8,9,10] and groundwater [11,12,13], leading to eutrophication pollution and nitrate nitrogen pollution in groundwater. Most of the sloping land with shallow soil layers is located in the upper reaches of the river, and the pollutant discharge from sloping land affects the health of the downstream population [14,15]. Nitrogen is highly mobile in soil and prone to loss via leaching and runoff, especially under high rainfall or excessive irrigation. These losses reduce nitrogen use efficiency, increase production costs [16], and contribute to severe environmental issues. Therefore, the assessment of nitrogen losses is necessary.

Irrigation is an effective way to achieve high and stable grain yield. However, most of the sloping land is located in economically underdeveloped areas, where rainfed agriculture predominates, often lacking irrigation facilities [17]. Because drought is more severe, the yield-increasing effect of irrigation in sloping farmland may be greater than that in plain areas. Thus, identifying the yield-enhancing potential of irrigation on sloping land can inform feasibility assessments for constructing irrigation infrastructure. Meanwhile, irrigation may also exacerbate water and nutrient losses [18]. Irrigation increases soil moisture; the soil will be easily saturated after heavy rainfall, resulting in surface runoff and leaching [19]. If the yield and environmental benefits of irrigation on shallow sloping land can exceed the investment in the facilities, it would be very valuable to build irrigation facilities in this area.

There have been few studies on the effects of irrigation on crop yield promotion on sloping farmland. Most of the research has been conducted in plain areas. The studies show that the crop yield and water use efficiency would be greatly improved by adjusting the irrigation amount [20,21] and frequency [22], irrigation method [23], and fertilizer application [24]. However, the research on the impact of irrigation on grain production in sloping land is rare. Existing irrigation studies on sloping land mainly focus on irrigation methods [25,26,27], such as drip irrigation, and the irrigation was mostly applied to cash crops [28].

Studies on water and nutrient loss have primarily explored biochar application [29], tillage methods [30], optimized nitrogen fertilization [31], and straw return modes [9]. However, few studies about the effect of irrigation on water and nitrogen loss have been performed, and most were conducted on paddy fields [32] and none on sloping land. For example, Liang et al. [33] investigated that optimized nitrogen and irrigation management can reduce the nitrogen loss and environmental risk in the double rice cropping system of South China. There is a lack of long-term, field-based studies evaluating the combined impacts of irrigation on crop yield, water dynamics, and nitrogen loss in sloping land with shallow soils.

In this study, a five-year irrigation experiment on sloping land with shallow soils was conducted to systematically evaluate both the agronomic and environmental benefits of irrigation. The research objectives were (1) to identify the effect of irrigation on grain yield promotion, and (2) to identify the effect of irrigation on surface runoff, leaching, and nitrogen loss in sloping land with shallow soils. This work provides important economic and environmental references for the establishment of irrigation systems in sloping land in southwest China.

## 2. Results

### 2.1. Precipitation

The monthly precipitation from 2016 to 2021 at the experimental site is shown in Table 1. The annual precipitation in years 2016, 2017, 2018, 2019, 2020, and 2021 was 667, 626, 1103, 1047, 800, and 1065 mm, respectively. The precipitation in 2016 and 2017 was low, and the precipitation in 2018, 2019, and 2021 was high. There was less precipitation from November to next March, accounting for 7.3% of the annual precipitation. The precipitation occurred mainly from June to September, accounting for 73.1% of the annual precipitation.

### 2.2. Yield

Irrigation can greatly promote crop yield (Figure 1). These yield promotions were greater in winter crops compared with maize. The average yield of winter crops from 2017 to 2021 in treatments SI and FI was 48.6% and 109.2% higher than RF, respectively. The winter crop yield difference between RF, SI, and FI was significant every year. The yield promotion was greatest in 2018; the yield of FI was 269.8% higher than RF. This dramatic difference resulted from severe drought stress during the critical rapeseed growth period from November 2017 to March 2018, when precipitation levels were substantially lower than in other years. The average yield of maize from 2017 to 2021 in treatments SI and FI was 6.7% and 23.4% higher than RF, respectively. When the precipitation was less during the maize growing season in 2017 and 2021, irrigation greatly increased the summer maize yield; the yield of FI was 61.1% and 64.7% higher than RF in 2017 and 2021. When the precipitation was sufficient during the maize growing season in 2018 and 2019, the maize yields had no obvious difference between each treatment for a few irrigations. Generally, the annual crop yield in treatments SI and FI was 16.4% and 43.5% higher than RF.

### 2.3. Runoff

As Figure 2 shows, runoff always occurred from March to October, and mostly from June to September, which was consistent with the precipitation. The runoff was higher in 2018, 2019, and 2021 due to more precipitation than in 2017 and 2020. The surface runoff in RF was significantly higher than treatments SI and FI in 2017, 2018, and 2021. The leaching volume in RF was less than SI and FI, and no significant and consistent difference occurred between SI and FI most of the time.

The averaged annual runoff is shown in Figure 3. Annual surface runoff was 84.4 mm in RF, which was 46.2% and 52.8% higher than in SI (57.7 mm) and FI (55.3 mm), respectively. Surface runoff did not differ significantly between SI and FI. The annual water leaching was 175.9 mm in RF, which was 13.7% and 13.6% lower than the values of SI (203.9 mm) and FI (203.6 mm), but not significantly different between SI and FI. The surface runoff accounted for 24.4% of the total runoff on average. There was no significant difference in total runoff between each treatment.

### 2.4. Nitrogen Loss

Due to the aerobic environment, soil ammonium nitrogen is converted to nitrate nitrogen through nitrification, resulting in low soil ammonium nitrogen concentrations. Furthermore, crop uptake and adsorption of ammonium ions (cations) by soil colloids limit their mobility. Therefore, the measured ammonium nitrogen concentration in water samples was consistently below 0.1 mg·L^−1^. The nitrate nitrogen concentration in the water samples was consistent with the total nitrogen (TN) concentration, indicating that nitrate nitrogen is the primary form of nitrogen loss. The TN concentration in surface runoff was always less than 3 mg L^−1^ and showed no significant differences in different treatments. In contrast, TN concentrations in leaching water were much higher and exhibited clear treatment effects; therefore, only the data of TN concentration in leaching water are shown in this paper (Figure 4).

The TN concentration in leaching water was increased after nitrogen fertilizer application. Over the five years, leaching samples were collected 32 times in total; the RF had a higher TN concentration than the irrigated treatments 23 times (significant 21 times), from 3 September 2017 to 23 October 2017, 11 May 2018 to 13 July 2018, 30 May 2019 to 7 July 2019, 28 July 2020 to 21 October 2020, and 2 July 2021 to 30 October 2021. The FI treatment had a lower TN concentration than other treatments 26 times (significant 23 times). Sometimes, the TN concentration differences between treatments were small and statistically insignificant, such as the TN concentration of the leaching water sampled from 15 April 2017 to 12 August 2017, and 17 August 2018 to 6 September 2018. Overall, it can be concluded that the TN concentration in leaching water decreased with irrigation in most cases.

The annual nitrogen loss is shown in Figure 5. The nitrogen loss through surface runoff ranged from 0.07 to 2.9 kg ha^−1^ per year, and the nitrogen loss through leaching water ranged from 4.8 to 64.7 kg ha^−1^ per year. Leaching was the main cause of nitrogen loss; 96% of the nitrogen loss was through the leaching water, and 4% through the surface runoff. There was no consistent difference in the nitrogen loss in surface runoff between different treatments from 2017 to 2020. The irritation had no significant effect on nitrogen loss through surface runoff according to the averaged five-year values. RF had higher nitrogen loss, and FI had lower nitrogen loss through leaching in 2017, 2018, 2020 and 2021. However, the nitrogen loss through leaching in 2019 was the opposite. The average annual total nitrogen loss through runoff was 35.4, 30.5, and 22.0 kgN ha^−1^ in RF, SI, and FI, respectively. Generally, nitrogen loss through leaching decreased with the irrigation, and the total nitrogen loss (including surface runoff and leaching) had the same results because leaching accounted for most of the nitrogen loss.

## 3. Discussion

### 3.1. Effect of Irrigation on Crop Yield

The full irrigation treatment (FI) promoted the yield of winter crops by 109.2% and increased the maize yield by 23.4% in this study. In contrast, Song et al. [22] conducted a four-year maize irrigation experiment in northeast China, and the full irrigation treatment increased the yield by 14–409% compared with the rainfed treatment, with an average yield increase of 67%. Mukherjee et al. [34] planted chickpea in the rice-fallows of the lower Gagetic plains of India in the dry season. The two-year average yield of the treatments irrigated at both flowering and pod periods was 107.8% higher than that of the rainfed treatment. Apparently, irrigation greatly promoted crop yield in sloping land with shallow soils, especially in the dry season.

The soil layer was shallow, and the soil water storage capacity of the profile was limited. Winter crops grew in the dry season with low precipitation; the limited soil water storage cannot satisfy the water demand by the crop, so irrigation can greatly promote the yield of winter crops. Although the precipitation was high in the maize growth period, soil storage can also be exhausted rapidly when more than ten days of high temperature and dry weather continue, and the maize yield is affected. The summer maize did not suffer drought stress in 2018 and 2019 because the precipitation was evenly distributed. Full irrigation can increase summer maize yield by 64.7% in years with insufficient precipitation.

The results showed that irrigation can promote crop yield in the shallow soil of southwest China. The annual yield increment between irrigation treatments and RF treatments was calculated, and the annual yield increment of the irrigation treatments had a significantly positive linear relationship with irrigation amount (Figure 6). On average, full irrigation can increase annual crop yield by 43.5%. According to the kernel prices (average price of each crop in each year during the harvest month, with reference to futures prices), winter wheat was 2.59 CNY/kg, rapeseed was 5.02 CNY/kg, maize was 2.07 CNY/kg, and the annual income increased by 11,294 CNY ha^−1^, equivalent to 1686 USD ha^−1^(average exchange rate from 2016 to 2021: 6.7). China government was promoting the construction of high-standard farmland in the experimental area, providing farmers with basic water conservancy facilities free of charge. On this basis, the cost for farmers to build a farmland irrigation system was about 30,000 CNY ha^−1^ (valid for more than 10 years), and the annual material and irrigation costs were 7500 CNY ha^−1^. The net profit was 794 CNY ha^−1^. Therefore, implementing irrigation on sloping land with shallow soil in southwest China can ensure food security and have great economic benefits.

### 3.2. Effect of Irrigation on Runoff

It is reasonable to expect that irrigation increases the soil moisture; the soil is more likely to saturate after rainfall, and more surface runoff and leaching may occur [18]. However, the irrigation did not increase the total runoff, but changed the distribution of surface runoff and leaching in the sloping land with shallow soil in this study. Irrigation decreased the surface runoff and increased the leaching. Why are these results different from imagination? First, irrigation was implemented when the crops were under drought stress. The winter crops were always irrigated between December and April of the next year, which was in the dry season with little precipitation. Irrigation increased the soil moisture, but the crops would gradually absorb this water. Meanwhile, runoff rarely occurred because of the lack of high-intensity rainfall during dry seasons. Although the maize grew in the rainy season, the maize was also irrigated when drought stress occurred and the soil was short of water storage. Because drought is usually persistent, there will be a high probability of continuous absence of rainfall following maize irrigation. The plants absorbed the irrigation water, and the soil moisture of the irrigated treatment dropped to the level of the RF treatment in a short period of time; after that, there will be no significant difference in the runoff between the irrigation treatment and the RF treatment when heavy rainfall occurs. In fact, if heavy rainfall occurred immediately after irrigation, the irrigation treatments would have more surface runoff and leaching than the RF treatment. This only occurred twice in the five-year experiment, after 14 April 2019 and 9 June 2019 irrigation, resulting in the total runoff in the irrigation treatment being higher than the RF treatment in the year 2019. However, these cases were of small probability. Second, irrigation can greatly promote crop growth and increase biomass and leaf area [35]. The increase in leaf area increases the crop coverage, which can effectively prevent raindrops from hitting the surface soil. The sloping land soil lacked soil organic matter and soil aggregates. When the raindrops hit the soil surface directly, they disperse the clay particles in the soil, block the soil pores, form a crust on the surface, and finally, decrease the infiltration of water [36]. Compared with irrigation treatments, the RF treatments had lower leaf area and crop coverage; therefore, surface runoff would easily form when the precipitation intensity was higher than the water infiltration. The rainfall was not fully infiltrated in the RF treatment, with the result that the leaching of RF treatment was less than the irrigation treatments. Third, higher plant biomass and leaf area meant higher water demand and uptake [37]; the irrigation water was offset by the increased water consumption, and the total runoff was not increased with irrigation.

Generally, irrigation increases soil moisture; it can also increase the risk of water runoff in some cases. However, irrigation promoted crop growth, improved crop coverage, and effectively reduced surface runoff. More soil water was consumed due to higher crop biomass and leaf area, and the leaching was reduced. Thus, appropriate irrigation did not increase runoff water loss in the sloping land with shallow soils in southwest China.

### 3.3. Effect of Irrigation on Nitrogen Loss

In this research, 96% of the nitrogen runoff loss was through leaching. Consistent with other research, leaching was the main cause of nitrogen runoff loss in sloping land [15]. The surface runoff flowed away quickly and could not be fully mixed with the soil, so the nitrogen concentration in the surface runoff was very low, with low nitrogen loss in surface runoff on sloping land [29].

The nitrogen loss through leaching decreased with the irrigation in 2017, 2018, 2020, and 2021. The leaching nitrogen loss was the product of the leaching water volume and nitrogen concentration of the water solution. The leaching water loss slightly increased with irrigation, so the reduction of the leaching nitrogen loss in irrigation treatments was mainly due to the lower nitrogen concentration of the leaching water solution. The crops grew better with more biomass accumulation, and roots absorbed more nutrients from the soil in the irrigation treatment [38]. The nitrogen fertilizer application was the same in all of the treatments; more nitrogen was absorbed by the crops, and less was left in the soil in the irrigation treatments, leading to the lower nitrogen concentration in the leaching water solution in the irrigation treatments. In 2019, heavy rainfall occurred immediately after irrigation on 14 April 2019 and 9 June 2019. This caused higher leaching, and fertilizer nitrogen had insufficient time to be absorbed by plants, resulting in greater nitrogen losses through leaching. During this period, the nitrogen losses of RF, SI, and FI were 14.0, 18.8, and 22.2 kg ha^−1^, respectively, accounting for 68.1%, 76.1%, and 80.0% of the total nitrogen loss in 2019. This highlights that irrigation can increase nitrogen loss under specific, infrequent circumstances.

The five-year average of yearly total nitrogen loss in full irrigation treatment (FI) was 22.0 kg ha^−1^, which decreased by 37.9% compared with the rainfed treatment (RF). Irrigation not only brought yield gains but also environmental benefits. Some readers may ask the question: If rainfed treatment has low biomass and less nitrogen requirements, why not reduce the application of nitrogen fertilizers, which can reduce the nitrogen loss by reducing the nitrogen concentration in the soil? This suggestion is not acceptable in crop production. The crop growth and yield greatly fluctuate in rainfed agriculture. It is difficult to accurately predict whether the crop will suffer drought stress in the future when fertilizing. If the amount of fertilizer application is reduced rashly, the crop yield will be decreased due to a lack of nutrients in the years without drought stress.

Appropriate irrigation can efficiently reduce nitrogen loss [39]. In this study, even in the full irrigation treatment, the irrigation volume was strictly controlled while the soil moisture was below the field capacity, and no runoff occurred. In fact, because of farmers’ lack of soil water knowledge, over-irrigation was common in agriculture production, which would increase the risk of nitrogen loss [18,40]. Therefore, appropriate irrigation can greatly reduce the nitrogen loss in purple sloping land.

## 4. Materials and Methods

### 4.1. Experimental Site

The experimental site is located at Liuma Village, Ziyang City, Sichuan Province, in the southwest of China (E 104°36′, N 30°7′). The altitude is 412 m. The climate type is a subtropical monsoon climate. The average annual temperature is 16.8 °C. The annual precipitation is 938.6 mm, and 70% falls between June and September. The terrain is hilly. The soil, locally known as purple soil, is classified as Regosols (RG) (WRB 2022 system). In the middle and top areas of the hills, the average thickness of the soil layer above the bedrock is only about 60 cm.

### 4.2. Experimental Facilities and Design

The experiments were conducted in concrete lysimeter plots with 4 m length and 2 m width and a slope of 8°. The bottom of the plot was covered with a 5 cm sand layer, and a drain was set at the lowest position. The plots were filled with a 60 cm depth of soil, which was consistent with the average soil thickness of the sloping cropland in the purple soil area. The lysimeter plot for this experiment was built in 2005. The soil in the lysimeter plots was the cultivated soil from the construction site, which was farmland before construction. At the beginning of each new test cycle, the soil in the plot was cleaned, and a new waterproof sheet was laid to ensure that the leaching water only flowed out from the drainage outlet at the bottom of the plot during the 3–5 year test cycle. The soil was fully mixed before filling and had the same soil properties, which had an organic C content of 5.3 g kg^−1^ (Walkley–Black method), total N of 0.71 g kg^−1^ (Kjeldahl digestion method), available P of 11.3 mg kg^−1^ (Olsen-P), and available K of 135.8 mg kg^−1^ (extracted with 1 mol L^−1^ ammonium acetate and quantified by flame photometry). The soil texture was silty clay loam with 19.7% sand, 50.3% silt, and 30.0% clay (pipette method). The 0–20 cm soil bulk density was 1.4 g cm^−3^, the soil retention curve parameters 0.1 Bar water content (tension table methods), wilting point (pressure plate apparatus) and residue water content (drying method) was 0.3 and 0.181 and 0.067 cm^3^ cm^−3^, and the stable water infiltration rate was 18 cm h^−1^ (double ring method). A collection trench at the front of each plot collected surface runoff. The leaching water was collected through the drain at the plot’s bottom. Surface runoff and leaching were stored in separate 500 and 300 L plastic containers with volume scales.

Three treatments were set in the experiment: rainfed control without irrigation (RF), single irrigation at flowering stage (SI), and full irrigation (FI). Every treatment had three replications. As Table 2 shows, the irrigation time and amount are inconsistent in different years. The SI treatment was irrigated in the flowering stage only when drought occurred (slight wilting of leaves indicates the onset of drought), and not irrigated if drought did not occur. The FI treatment was irrigated whenever the crop experienced drought stress.

Irrigation methods and amounts were optimized annually, as follows:

From 12 November 2016 to 8 June 2017, 30 mm of water was applied during each drought period using movable pipes. To avoid inducing surface runoff with manual watering, irrigation was split into two applications per event (15 mm).

On 30 June 2017, sprinkler irrigation equipment was introduced to apply 30 mm in a single event.

Recognizing that 30 mm per irrigation might be insufficient and lead to excessive frequency, from 2018 to 2020, the FI irrigation amount was calculated based on the profile water deficit (profile field water capacity minus actual profile water storage). The SI irrigation amount was set equal to the FI amount during its single application.

In 2021, considering SI had only one irrigation opportunity and the soil was likely drier, the irrigation amount for both SI and FI was calculated based on their respective profile water deficits, potentially differing between treatments.

### 4.3. Field Management

The cropping system was winter wheat (*Triticum aestivum* L.) and summer maize (*Zea mays* L.) rotation from 2016 to 2017 to facilitate mechanized production. Because rapeseed is the main crop in the experimental area and the winter wheat in the experimental plots suffered severe bird damage in 2017, the cropping system was adjusted to rapeseed (*Brassica napus* L.) and summer maize rotation from 2017 to 2021. The wheat and maize sowing and rapeseed transplanting date, crop harvest, and fertilizer date are shown in Table 3. The fertilizer application of winter wheat and rapeseed was N 180 kg ha^−1^, P_2_O_5_ 90 kg ha^−1^, and K_2_O 75 kg ha^−1^. The fertilizer application of maize was 225 kg ha^−1^, P_2_O_5_ 120 kg ha^−1^, and K_2_O 75 kg ha^−1^. The nitrogen fertilizer is urea, the phosphorus fertilizer is superphosphate, and the potassium fertilizer is potassium chloride. A total of 60% of the N fertilizer for winter wheat and rapeseed was applied as base fertilizer, and the other 40% was top dressing. A total of 40% of the N fertilizer for maize was applied as base fertilizer, and the other 60% was top dressing. All the P and K were applied as base fertilizer before sowing. The topdressing time was the jointing stage of winter wheat, the bolting stage of rapeseed, and the tasseling stage of maize. The fertilizer was applied to the soil surface around the plants. The soil was tilled before planting winter crops, but not tilled before planting maize. The crop residue was removed from the field. The winter wheat cultivar was Chuanmai104, the rapeseed cultivar was Yuyou 21, and the summer maize cultivar was Zhenghong 6. Other cultivation parameters followed local farmer practices.

### 4.4. Sampling and Measurement

When heavy rainfall produced surface runoff and leaching, the volume of liquid in the collection containers was recorded. From the fully fixed liquid in the containers, 100 mL water sampler samples were collected, which were stored at 4 °C and analyzed within 48 h. The containers were then emptied. The runoff monitoring and sampling continued after the next runoff occurred.

The ammonium nitrogen, nitrate nitrogen, and total nitrogen concentrations of the water samples were measured. Ammonium nitrogen measurement: Mix the water sample with hypochlorite and phenol solution; the ammonium ions will react to produce indophenol blue, and the ammonium concentration is determined by colorimetry. Nitrate nitrogen measurement: After the water sample was filtered, the concentration of nitrate was directly measured with an ultraviolet spectrophotometer. Total nitrogen (TN) measurement: Add alkaline potassium persulfate oxidant to the water sample, oxidize all nitrogen to nitrate nitrogen through high temperature, then filter the solution and measure the concentration of nitrate ions with an ultraviolet spectrophotometer.

The water and nitrogen loss in a year or in one crop growth season was accumulated by the water and nitrogen loss of each surface runoff and leaching. The nitrogen loss of each runoff was the product of runoff volume and TN concentration. The ears of maize in the plot were manually harvested when the crop was mature. The kernels were air-dried with a moisture content of 14% and weighed to determine the yield.

Precipitation was monitored using a weather station adjacent to the experimental site.

### 4.5. Statistical Analysis

The statistically significant differences between each treatment were determined by the Least significant difference (LSD) test at *p* = 0.05. The linear regression method was used to describe the relationship between variables. Statistical analyses were performed using SPSS 17.0.

## 5. Conclusions

This study evaluated the effect of irrigation on crop production and water and nitrogen loss in sloping land with shallow soil in southwest China. Appropriate irrigation greatly promoted crop production; the annual yield in SI and FI was 16.4% and 43.5% higher than rainfed treatment (RF), with an economic benefit of 1912 USD ha^−1^. Irrigation decreased surface runoff but increased leaching, resulting in no net impact on total runoff. A total of 96% of the nitrogen runoff loss was through leaching. Irrigation significantly increased crop production and nitrogen uptake, thereby decreasing nitrogen concentration in the leaching and ultimately reducing nitrogen leaching loss. The annual nitrogen loss in FI was decreased by 37.8% compared with RF. Thus, appropriate irrigation on sloping land with shallow soil had both economic and environmental benefits.

## Figures and Tables

**Figure 1 plants-14-02666-f001:**
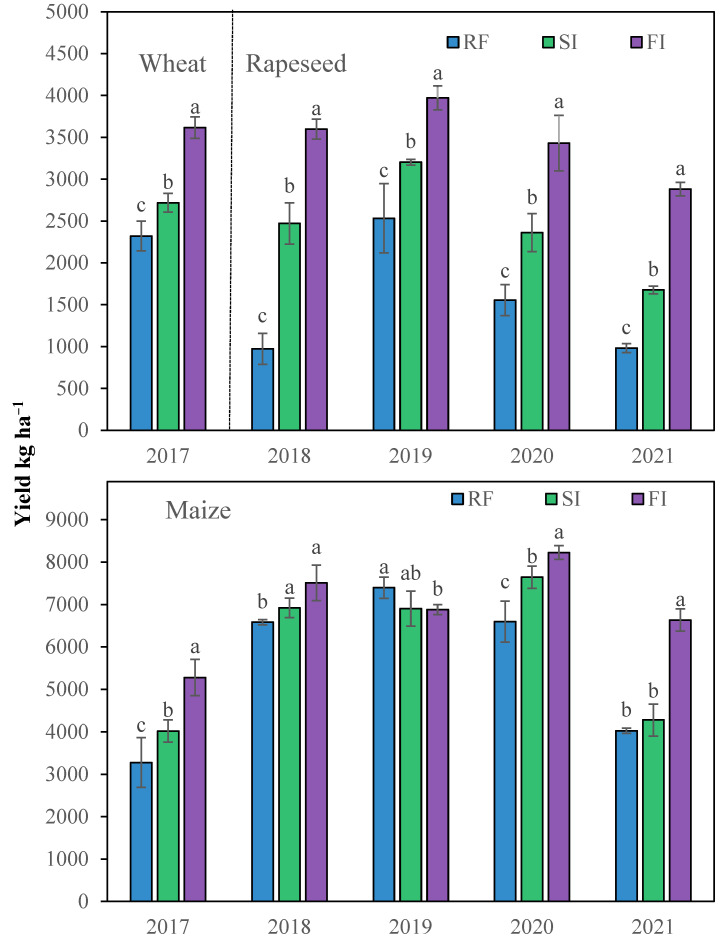
Yield of winter wheat, rapeseed, and maize under different irrigated treatments from 2017 to 2021. Note: RF, rainfed; SI, single irrigation; FI, full irrigation. Different letters in the same year indicate significant difference in the mean value by LSD test, *p* < 0.05.

**Figure 2 plants-14-02666-f002:**
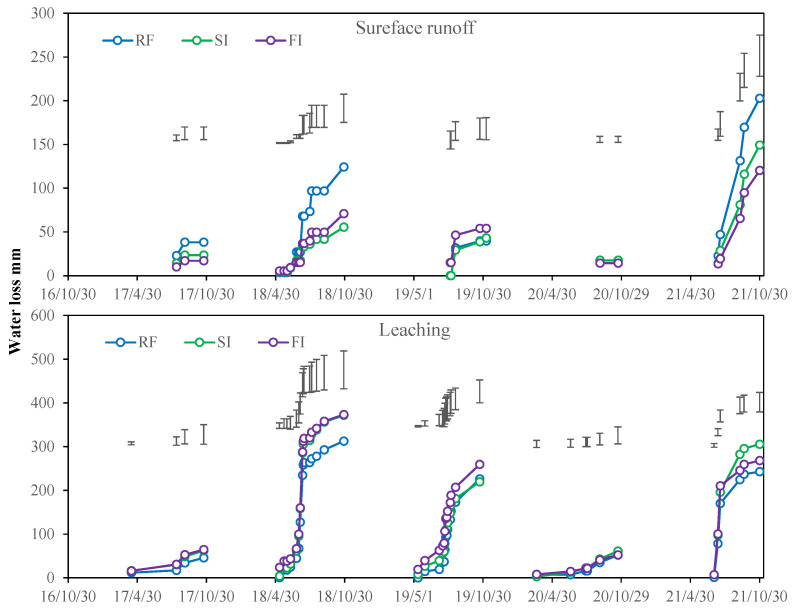
Accumulation of surface runoff and leaching in different irrigation treatments over the years from 2016 to 2021. Note: RF, rainfed; SI, single irrigation; FI, full irrigation. Vertical bars represent the LSD value at *p* = 0.05 (*n* = 3).

**Figure 3 plants-14-02666-f003:**
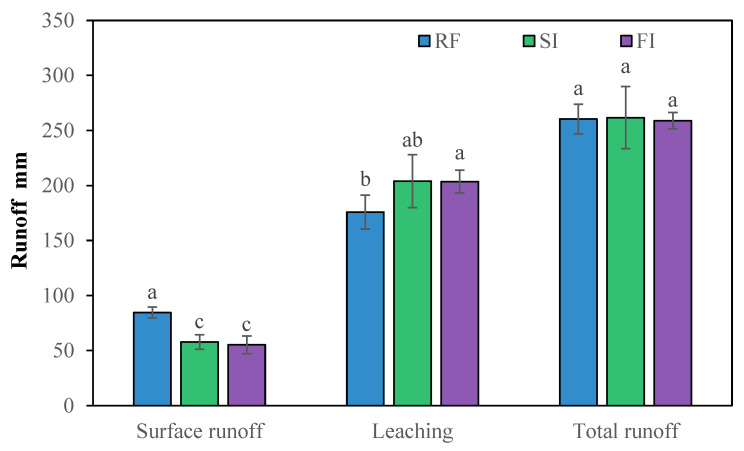
Averaged annual surface runoff, leaching, and total runoff at different irrigated treatments. Note: RF, rainfed; SI, single irrigation; FI, full irrigation. Different letters indicate significant difference in the mean value by LSD test, *p* < 0.05.

**Figure 4 plants-14-02666-f004:**
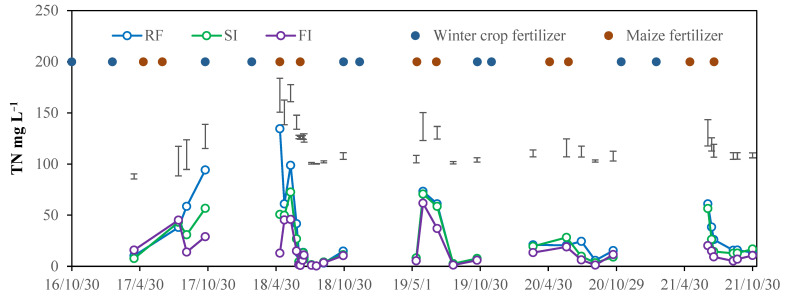
Dynamic of TN concentration in each leaching water in different irrigated treatments from 2016 to 2021. Note: RF, rainfed; SI, single irrigation; FI, full irrigation. The spot means the fertilizer time. Vertical bars represent the LSD value at *p* = 0.05 (*n* = 3).

**Figure 5 plants-14-02666-f005:**
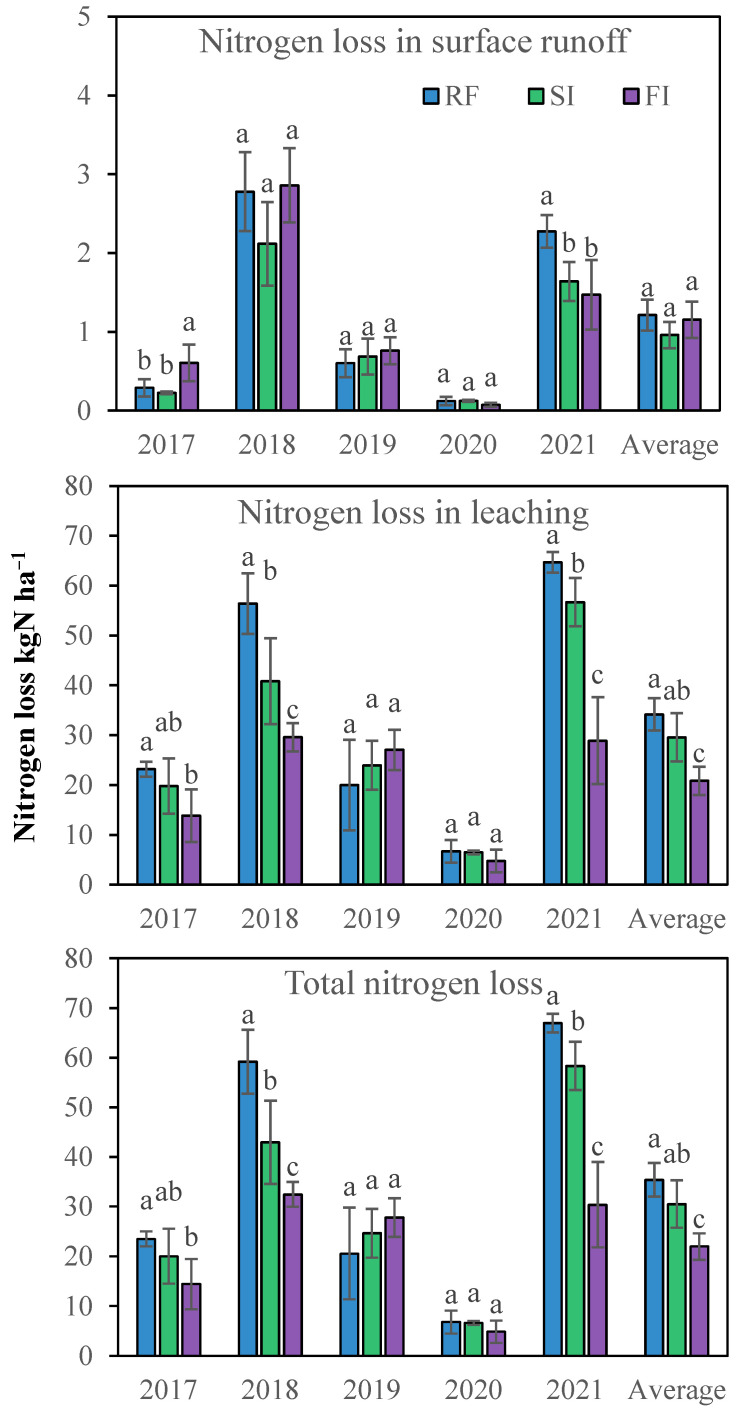
Averaged annual nitrogen loss in surface runoff, leaching water, and total runoff in different irrigated treatments from 2017 to 2021. Note: RF, rainfed; SI, single irrigation; FI, full irrigation. Different letters in the same year indicate significant difference in the mean value by LSD test, *p* < 0.05.

**Figure 6 plants-14-02666-f006:**
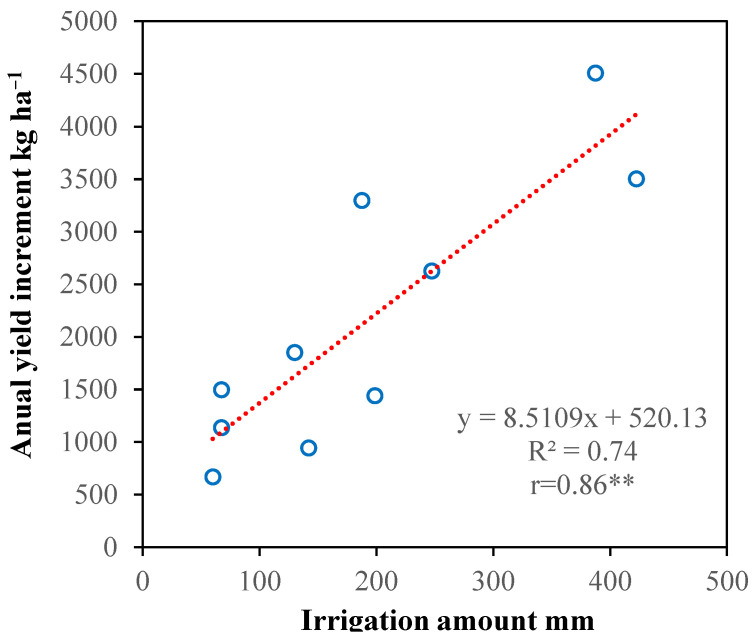
The relationship between annual yield increment and yearly irrigation amount from 2016 to 2021. The annual yield increment is the yield increment of the irrigated treatments to the rainfed treatment. Linear regression method was used to describe the relationship. ** indicates a significant positive correlation (*p* < 0.01).

**Table 1 plants-14-02666-t001:** Monthly precipitation (mm) from 2016 to 2021.

Year	Jan	Feb	Mar	Apr	May	Jun	Jul	Aug	Sep	Oct	Nov	Dec	Total
2016	14	39	16	46	76	133	132	57	119	19	11	6	667
2017	5	4	29	21	38	78	174	176	39	54	6	4	626
2018	6	4	26	72	103	212	323	145	122	59	11	21	1103
2019	2	19	25	21	138	98	312	204	133	63	30	2	1047
2020	8	6	33	27	36	134	135	235	106	49	8	23	800
2021	14	9	5	28	77	81	319	252	228	42	6	4	1065

**Table 2 plants-14-02666-t002:** Irrigation date and amount (mm) for winter wheat, rapeseed, and maize from 2016 to 2021.

Date	Crops	RF	SI	FI	Irrigation Methods
22 November 2016	winter wheat	/	/	15	Moveable Pipe Irrigation
4 December 2016	winter wheat	/	/	15	Moveable Pipe Irrigation
12 February 2017	winter wheat	/	/	15	Moveable Pipe Irrigation
24 February 2017	winter wheat	/	/	15	Moveable Pipe Irrigation
4 March 2017	winter wheat	/	15	15	Moveable Pipe Irrigation
21 March 2017	winter wheat	/	15	15	Moveable Pipe Irrigation
11 May 2017	maize	/	/	15	Moveable Pipe Irrigation
14 May 2017	maize	/	/	15	Moveable Pipe Irrigation
7 June 2017	maize	/	/	15	Moveable Pipe Irrigation
8 June 2017	maize	/	/	15	Moveable Pipe Irrigation
30 June 2017	maize	/	30	30	Springer
13 January 2018	rapeseed	/	/	67.5	Springer
1 March 2018	rapeseed	/	67.5	67.5	Springer
21 March 2018	rapeseed	/	/	45	Springer
16 April 2018	rapeseed	/	/	67.5	Springer
27 December 2018	rapeseed	/		52.5	Springer
26 March 2019	rapeseed	/	60	60	Springer
14 April 2019	rapeseed	/	/	60	Springer
9 June 2019	maize	/	/	26	Springer
25 December 2019	rapeseed	/	/	45	Micro springer
29 February 2020	rapeseed	/	60	60	Micro springer
12 March 2020	rapeseed	/	/	60	Micro springer
25 March 2020	rapeseed	/	/	60	Micro springer
5 April 2020	rapeseed	/	/	60	Micro springer
6 May 2020	maize	/	/	68	Micro springer
11 June 2020	maize	/	60	60	Micro springer
5 February 2021	rapeseed	/	/	26.7	Micro springer
22 February 2021	rapeseed	/	/	26.7	Micro springer
8 March 2021	rapeseed	/	/	40	Micro springer
18 March 2021	rapeseed	/	94.7	93.3	Micro springer
13 April 2021	rapeseed	/	/	80	Micro springer
21 June 2021	maize	/	40	26.7	Micro springer
29 July 2021	maize	/	/	46.7	Micro springer
11 August 2021	maize	/	/	40	Micro springer

Note: RF, rainfed; SI, single irrigation; FI, full irrigation; “/” means no irrigation.

**Table 3 plants-14-02666-t003:** The date of crop sowing, transplanting, harvest, and fertilizer application.

Field Management	2016–2017	2017–2018	2018–2019	2019–2020	2020–2021
Winter crop sowing or transplanting	9 November 2016	14 October 2017	25 October 2018	14 October 2019	4 November 2020
Winter crop base fertilizer	30 October 2016	23 October 2017	29 October 2018	23 October 2019	12 November 2020
Winter crop top dressing	16 February 2017	25 February 2018	11 December 2018	30 November 2019	14 February 2021
Winter crop harvest	3 May 2017	2 May 2018	8 May 2019	26 April 2020	9 May 2021
Summer maize sowing and base fertilizer	10 May 2017	11 May 2018	14 May 2019	4 May 2020	15 May 2021
Summer maize top dressing	30 June 2017	5 July 2018	5 July 2019	23 June 2020	19 July 2021
Summer maize harvest	30 August 2017	25 August 2018	21 August 2019	2 September 2020	24 August 2021

## Data Availability

The data are accessible within the article.

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
