# Peer review of "Effect of Irrigation on Crop Yield and Nitrogen Loss in Simulated Sloping Land with Shallow Soils"

_plants, 2025, doi:10.3390/plants14172666_

Round 1

Reviewer 1 Report (New Reviewer)

Comments and Suggestions for Authors

Upon reviewing the manuscript titled “Effect of Irrigation on Crop Yield and Nitrogen Loss in Simulated Sloping Land with Shallow Soils,” I noticed that the red-colored text indicates the authors have submitted a revised version and have addressed previous comments. The study includes three treatments: rainfed control (RF), single irrigation at the flowering stage (SI), and full irrigation (FI), aiming to investigate the effects of irrigation on surface runoff, leaching, and nitrogen loss in sloping land with shallow soils.

  1. The authors used three crops—winter wheat, rapeseed, and maize—yet the experimental design lacks details regarding how each crop was incorporated into the study. Please clarify this in the Materials and Methods section.

  1. Line 10–13: Please revise the sentence with the following improved version for clarity and conciseness: “Although irrigation is a proven strategy to mitigate drought stress and enhance yields, increased soil moisture under irrigation may exacerbate water and nitrogen losses. Therefore, this study aimed to investigate the long-term effects of irrigation regimes on crop yield, surface runoff, leaching, and nitrogen loss in shallow soil systems.”
  2. The introduction lacks a clear and logical structure. Key ideas—such as water/nutrient loss and drought vulnerability of sloping land—are repeated and not grouped into coherent thematic paragraphs. The section jumps between environmental concerns, crop yield potential, irrigation feasibility, and research gaps without appropriate transitions. A thorough restructuring is necessary. Please organize the introduction as follows: Importance of the study topic, Supporting evidence and background, Statement of the problem with relevant citations, Review of previous approaches and existing research gaps, Explanation of how the present study addresses the problem, Clearly stated aim and objectives, A one-sentence hypothesis.
  3. Also, the importance of nitrogen is not sufficiently discussed. Please include at least two to three sentences explaining its role in crop productivity and environmental impact, preferably citing recent literature (e.g., https://doi.org/10.56946/jspae.v4i2.661).
  4. Line 51: The sentence “Does irrigation increase the risk of water and nutrient loss in sloping land? Yes,” is redundant and disrupts the flow. Please revise this and improve the structure of the entire introduction accordingly.
  5. Line 93–102 (Section 2.1): The section of precipitation belongs in the Materials and Methods section under "Experimental Site Description," not in the Results section.
  6. Lines 103–117: The results section discusses yield but does not specify which crop is being referred to. As three different crops were used in different years, this can confuse readers. Please specify crop names and clarify yield data accordingly.
  7. Line 196–221 (Discussion Section): The discussion contains unnecessary background repetition. Since the importance and problems are already introduced earlier, this section should begin with the current study's findings, followed by brief explanations. Then, compare and contrast your results with previous studies (e.g., Line 196–221). This approach will improve clarity and avoid redundancy.
  8. Figure 7: It would be more helpful to provide a clear, labeled image showing the slope and crop layout. If not possible, consider moving this figure to the Supplementary Materials, as it is not essential in the main text under Materials and Methods.
  9. Lines 414–418: The use of one-way ANOVA is unclear, considering that the study involves multiple treatments and multiple years. Please clarify the statistical model used and justify its appropriateness for this experimental design.

Author Response

Comment 1: The authors used three crops—winter wheat, rapeseed, and maize—yet the experimental design lacks details regarding how each crop was incorporated into the study. Please clarify this in the Materials and Methods section.

Response: Thank you for the comment. The original experimental design specified a winter wheat–summer maize rotation to facilitate mechanized production. However, in practice, local farmers in the study area predominantly cultivate rapeseed in winter. In addition, the winter wheat in our experimental plots suffered severe bird damage during the first season. As a result, the cropping system was adjusted to a rapeseed–summer maize rotation. This information has been clarified in the revised manuscript (Lines 373–375).

Comment 2: Line 10–13: Please revise the sentence with the following improved version for clarity and conciseness: “Although irrigation is a proven strategy to mitigate drought stress and enhance yields, increased soil moisture under irrigation may exacerbate water and nitrogen losses. Therefore, this study aimed to investigate the long-term effects of irrigation regimes on crop yield, surface runoff, leaching, and nitrogen loss in shallow soil systems.”

Response: Thanks for the comments. we have revised it in Line 12-16.

Comment 3: The introduction lacks a clear and logical structure. Key ideas—such as water/nutrient loss and drought vulnerability of sloping land—are repeated and not grouped into coherent thematic paragraphs. The section jumps between environmental concerns, crop yield potential, irrigation feasibility, and research gaps without appropriate transitions. A thorough restructuring is necessary. Please organize the introduction as follows: Importance of the study topic, Supporting evidence and background, Statement of the problem with relevant citations, Review of previous approaches and existing research gaps, Explanation of how the present study addresses the problem, Clearly stated aim and objectives, A one-sentence hypothesis.

Response: Thank you for the comment. We have reorganized the introduction section, according to the suggestions above.

Comment 4: Also, the importance of nitrogen is not sufficiently discussed. Please include at least two to three sentences explaining its role in crop productivity and environmental impact, preferably citing recent literature (e.g., https://doi.org/10.56946/jspae.v4i2.661).

Response: Thank you for the comment. it is necessary to explain the impact of nitrogen on crop productivity and environmental. We add “Nitrogen is highly mobile in soil and prone to loss via leaching and runoff, especially under high rainfall or excessive irrigation. These losses reduce nitrogen use efficiency, increase production costs [16], and contribute to severe environmental issues. Therefore, the assessment of nitrogen losses is necessary.” in Lines 53–56.

Comment 5: Line 51: The sentence “Does irrigation increase the risk of water and nutrient loss in sloping land? Yes,” is redundant and disrupts the flow. Please revise this and improve the structure of the entire introduction accordingly.

Response: Thank you for the comment. We have made significant changes to the introduction and modified the expression.

Comment 6: Line 93–102 (Section 2.1): The section of precipitation belongs in the Materials and Methods section under "Experimental Site Description," not in the Results section.

Response: Thank you for the comment. The rainfall data presented in this section are not long-term annual averages but detailed monthly data collected over six consecutive years. Because precipitation patterns directly influence both crop yield and runoff dynamics in our experiment, we consider these data to be an integral part of the Results. For this reason, we have retained them in the Results section rather than moving them entirely to the “Experimental Site Description”.

Comment 7: Lines 103–117: The results section discusses yield but does not specify which crop is being referred to. As three different crops were used in different years, this can confuse readers. Please specify crop names and clarify yield data accordingly.

Response: Thank you for the comment. In this study, winter wheat was cultivated as the winter crop for one year, rapeseed for four years, and maize served as the summer crop throughout. In the Results section, yields of winter crops and summer maize were analyzed separately. Because winter wheat was grown only in the first year and its yield performance during its growing season was comparable to that of rapeseed, we combined wheat and rapeseed data for analysis. (Lines 105–115).

Comment 8: Line 196–221 (Discussion Section): The discussion contains unnecessary background repetition. Since the importance and problems are already introduced earlier, this section should begin with the current study's findings, followed by brief explanations. Then, compare and contrast your results with previous studies (e.g., Line 196–221). This approach will improve clarity and avoid redundancy.

Response: Thank you for the suggestion. In the second paragraph of Section 3.1, we provide a detailed explanation of why irrigation has a greater effect on winter crop yields than on summer maize yields. We acknowledge that this discussion partially overlaps with content in the Introduction. However, given that this paper focuses on evaluating yield benefits, our discussion of yield formation mechanisms is necessarily qualitative due to limitations in analytical methods and other factors. We have retained this explanation because it is important for helping non-specialist readers understand the practical implications of our findings.

Comment 9: Figure 7: It would be more helpful to provide a clear, labeled image showing the slope and crop layout. If not possible, consider moving this figure to the Supplementary Materials, as it is not essential in the main text under Materials and Methods.

Response: Thank you for your suggestion. Due to lack of preparation, we do not have a more attractive and illustrative photo, and the information shown in the original image is fully described in the main text. Therefore, we have removed this image

Comment 10: Lines 414–418: The use of one-way ANOVA is unclear, considering that the study involves multiple treatments and multiple years. Please clarify the statistical model used and justify its appropriateness for this experimental design.

Response: Thank you for your suggestion. Because the significant differences between irrigation treatments and years were so pronounced in this paper, we did not include a table of variance analysis results. We used the Least Significant Difference (LSD) method to test for significant differences between each treatment. For a more accurate description, we have removed the ANOVA description. (Line 416)

Reviewer 2 Report (New Reviewer)

Comments and Suggestions for Authors

Review

 In the manuscript entitled "Effect of Irrigation on Crop Yield and Nitrogen Loss in Simulated Sloping Land with Shallow Soils," the authors presented results regarding the impact of irrigation on yield and nitrogen losses. However, after analyzing the text, I believe it requires additions and corrections of errors and unclear wording. Below are my comments:

- The manuscript is written in a non-standard manner. In my opinion, for clarity, the standard order of chapters should be adhered to, namely, the material and methods section should appear immediately after the introduction, rather than following the discussion. Please change this.

 - Line 319: "The soil, locally known as purple soil, is classified as an Entisol."There is no information about the classification system for this soil.Furthermore, I believe the authors should present the classification of the analyzed soil according to the international WRB 2022 system. This would be understandable to a wider audience of readers dealing with this issue.

- The manuscript's subject line uses the phrase "Sloping land with shallow soils" – however, the reader is not provided with detailed information about this soil. There is no description of the soil profile or detailed taxonomy - please complete

- Line 149: "The ammonium nitrogen concentration of the water sample was always lower than 0.1 mg L-1 because of the aerobic environment of the sloping land soil, and the nitrate nitrogen concentration was the same as the total nitrogen concentration." I don't understand this sentence. Please explain what this means.

Furthermore, I believe that the remaining phrases in this chapter require correction – rewording.

Lines 332: "The soil was fully mixed before filling and had the same soil properties, which had an organic C content of 5.3 g kg−1, total N of 0.71 g kg−1, available P of 11.3 mg kg−1, and available K of 135.8 mg kg−1.The soil texture was silty clay loam with 19.7% sand, 50.3% silt, and 30.0% Clay.The soil retention curve parameters 0.1 Bar water content, wilting point and residue water content was 0.3 and 0.181 and 0.067 cm3 cm-3, and the stable water infiltration rate was 3 mm min-1” questions arise here: what method was used to extract available phosphorus and potassium, because this is the basis for the interpretation of the obtained values.Furthermore, I think it's better to report water content as a percentage of mass per volume (% m/V).However, the infiltration coefficient is better reported in international units – SI, i.e., m/s.I also have a question: what method was used to determine soil texture, as this is also helpful for interpreting the results?

- Line 357: "In 2021, considering SI had only one irrigation opportunity and the soil was likely drier, the irrigation amount for both SI and FI was calculated based on their respective profile water deficits, potentially differing between treatments."  To my knowledge, it is commonly assumed that irrigation begins when soil moisture drops below 60% of field water capacity. My question is why the authors adopted a different solution in their experiment, why tensiometers or another method of determining irrigation needs were not used.

In my opinion, before the manuscript is allowed to proceed to the next stages of the editorial process, it requires the authors to make corrections, clarify inconsistencies and remove errors.

Author Response

Comment 1: The manuscript is written in a non-standard manner. In my opinion, for clarity, the standard order of chapters should be adhered to, namely, the material and methods section should appear immediately after the introduction, rather than following the discussion. Please change this.

Response: Thank you for the comment. According to the magazine's requirements, methods and materials are placed at the end.

Comment 2: Line 319: "The soil, locally known as purple soil, is classified as an Entisol."There is no information about the classification system for this soil.Furthermore, I believe the authors should present the classification of the analyzed soil according to the international WRB 2022 system. This would be understandable to a wider audience of readers dealing with this issue.

Response: Thank you for the comment. We have changed soil type “Entisol” to “Regosols (RG)” according to the international WRB 2022 system (Lines 322).

Comment 3: The manuscript's subject line uses the phrase "Sloping land with shallow soils " – however, the reader is not provided with detailed information about this soil. There is no description of the soil profile or detailed taxonomy - please complete

Response: Thank you for the comment. We explained it in introduction. “The soil of sloping land in southwest China develops from purple sedimentary rock and limestone. Due to the serious soil erosion on sloping farmland, the soil layer is very shallow, 75% of which ranges from 15 to 75 cm.” Therefore, what we call shallow soil sloping farmland refers to this type of sloping land.” (Lines 42-44).

Comment 4: Line 149: "The ammonium nitrogen concentration of the water sample was always lower than 0.1 mg L-1 because of the aerobic environment of the sloping land soil, and the nitrate nitrogen concentration was the same as the total nitrogen concentration." I don't understand this sentence. Please explain what this means.

Response: Thank you for the comment. Because dryland soils are aerobic, ammonium nitrogen in the soil solution is converted to nitrate nitrogen through nitrification. Therefore, the measured ammonium nitrogen concentration in water samples was consistently below 0.1 mg·L⁻¹. The nitrate nitrogen concentration in the water samples was consistent with the total nitrogen concentration, indicating that nitrate nitrogen is the primary form of nitrogen loss. We revised the manuscript in Line 150-154.

Comment 5: Furthermore, I believe that the remaining phrases in this chapter require correction – rewording.

Response: Thank you for the comment. We reviewed the section 2.4 and improved the expression in line 160-181.

Comment 6: Lines 332: "The soil was fully mixed before filling and had the same soil properties, which had an organic C content of 5.3 g kg−1, total N of 0.71 g kg−1, available P of 11.3 mg kg−1, and available K of 135.8 mg kg−1.The soil texture was silty clay loam with 19.7% sand, 50.3% silt, and 30.0% Clay.The soil retention curve parameters 0.1 Bar water content, wilting point and residue water content was 0.3 and 0.181 and 0.067 cm3 cm-3, and the stable water infiltration rate was 3 mm min-1” questions arise here: what method was used to extract available phosphorus and potassium, because this is the basis for the interpretation of the obtained values. Furthermore, I think it's better to report water content as a percentage of mass per volume (% m/V).However, the infiltration coefficient is better reported in international units – SI, i.e., m/s.I also have a question: what method was used to determine soil texture, as this is also helpful for interpreting the results?

Response: Thank you for the comment. We have added the relevant measurement methods for available phosphorus, available potassium, and soil texture in the revised manuscript. Volumetric water content and soil water potential are used in soil retention curve, so the volumetric water content was used here. We added the bulk density which can used to calculate gravimetric water content. In addition, the infiltration rate units have been changed from mm·min⁻¹ to cm·h⁻¹, which is more commonly used and consistent with international practice (Lines 336-343).

Comment 7: Line 357: "In 2021, considering SI had only one irrigation opportunity and the soil was likely drier, the irrigation amount for both SI and FI was calculated based on their respective profile water deficits, potentially differing between treatments."  To my knowledge, it is commonly assumed that irrigation begins when soil moisture drops below 60% of field water capacity. My question is why the authors adopted a different solution in their experiment, why tensiometers or another method of determining irrigation needs were not used.

Response: Thank you for the comment. Thanks for the professional question. From the perspective of optimal water management, irrigation is undoubtedly the most reasonable approach when the moisture content is below 60% of field water capacity. However, implementing this strategy in actual production is challenging. The accuracy and installation depth of the moisture probe or tensiometer will affect the measurement results, which in turn affects the timing and amount of irrigation. We lack real-time monitoring equipment and experience in precision water management. Consequently, we adopted a simpler method: irrigation was triggered when the first signs of leaf wilting appeared, and the irrigation volume was calculated based on the soil moisture content at that moment.

Round 2

Reviewer 2 Report (New Reviewer)

Comments and Suggestions for Authors

Review

The authors submitted a revised version of the manuscript, largely in line with my comments and suggestions. I would only disagree with this point:

Comment 4: Line 149: "The ammonium nitrogen concentration of the water sample was always lower than 0.1 mg L-1 because of the aerobic environment of the sloping land soil, and the nitrate nitrogen concentration was the same as the total nitrogen concentration." I don't understand this sentence. Please explain what this means.

Response: Thank you for the comment. Because dryland soils are aerobic, ammonium nitrogen in the soil solution is converted to nitrate nitrogen through nitrification. Therefore, the measured ammonium nitrogen concentration in water samples was consistently below 0.1 mg·L⁻¹. The nitrate nitrogen concentration in the water samples was consistent with the total nitrogen concentration, indicating that nitrate nitrogen is the primary form of nitrogen loss. We revised the manuscript in Line 150-154.

my suggestion:

In my opinion, the differences in leaching result from the fact that nitrogen in the ammonium form, as a cation, can be sorbed interchangeably and therefore is subject to a low leaching rate. Nitrate nitrogen, as a readily water-soluble anion, does not possess these properties; if it is not biologically sorpted (taken up by living organisms), it is easily leached.

After minor changes, I recommend the manuscript for further editorial review.

Author Response

Comment 1: In my opinion, the differences in leaching result from the fact that nitrogen in the ammonium form, as a cation, can be sorbed interchangeably and therefore is subject to a low leaching rate. Nitrate nitrogen, as a readily water-soluble anion, does not possess these properties; if it is not biologically sorpted (taken up by living organisms), it is easily leached.

Response: Thanks. We fully agree with your comment. The result analyze was inadequate. In the aerobic environment, strong nitrification processes rapidly convert ammonium nitrogen (released from fertilizer and organic matter decomposition) into nitrate nitrogen. This process was under an equilibrium. In our historical soil sample analyses, ammonium nitrogen concentrations were typically ranged between 0–10 mg kg⁻¹ which were much lower than nitrate nitrogen. However, the ammonium nitrogen in soil leaching water were below 0.1 mg kg⁻¹ in this study, it was attributable to crop uptake and adsorption  by soil colloids.

We have revised the manuscript as follows: “Due to the aerobic environment, soil ammonium nitrogen is converted to nitrate nitrogen through nitrification, resulting in low soil ammonium nitrogen concentrations. Furthermore, crop uptake and adsorption of ammonium ions (cations) by soil colloids limit their mobility. Consequently, the ammonium nitrogen concentrations in the leaching water were less than 0.1 mg L⁻¹”. (Line 150-151)

This manuscript is a resubmission of an earlier submission. The following is a list of the peer review reports and author responses from that submission.

Round 1

Reviewer 1 Report

Comments and Suggestions for Authors

The manuscript presents a well-designed and insightful five-year study on the effects of irrigation on crop productivity and nitrogen loss in sloping land with shallow soils in Southwest China. The long-term dataset and lysimeter-based approach provide enough evidence that appropriate irrigation strategies can enhance crop yield while minimizing nitrogen leaching. The results are of both scientific and practical significance for sustainable land management in similar topographies. However, the manuscript would benefit from clearer figure presentation, improved language precision, and a few clarifications in the methods and interpretation of results.

Comments and Suggestions for Authors

  1. Page 2, Lines 48–51: The manuscript discusses the potential risk of increased runoff and leaching due to irrigation but does not explain clearly how this was mitigated in practice. Please expand on how the irrigation scheduling was aligned with crop water stress and rainfall forecasts to prevent these risks in the experimental design.
  1. Page 3, Lines 97–109: The discussion on yield increase between SI, FI, and RF treatments is informative, but the large range of values (e.g., “269.8% higher” in 2018) should be contextualized better. Please briefly explain why 2018 showed such extreme results. Was there a notable weather anomaly or irrigation difference? Adding a short explanation would help readers understand this variation.
  2. Page 6, Lines 148–158: The TN concentration trend in leaching water is discussed in a narrative manner, but the statistical significance is unclear. Please clarify whether the observed differences (23 out of 32 times) are statistically significant and consider presenting summary statistics in a table for easier interpretation. This would enhance the clarity of the nitrogen leaching trend over time.
  3. Page 8, Lines 210–212: The claim of economic benefit (1912 USD ha-1) from full irrigation is convincing, but it assumes a fixed market price. Please clarify whether these prices are averages across years, and mention any assumptions about irrigation cost (e.g., infrastructure, labor). A brief caution or sensitivity note would strengthen the argument’s credibility.
  4. Page 10, Lines 283–284: In the discussion of nitrogen loss in 2019 due to post-irrigation rainfall, the mechanism is well-explained. However, it would be beneficial to quantify how much this specific event contributed to the annual TN loss in that year compared to the average. Including a percentage or absolute difference would better illustrate the exceptionality of 2019.
  5. Page 11, Lines 311–324: The lysimeter system and collection method are well described. However, it is unclear whether there was any calibration or maintenance performed during the 5-year period to ensure measurement consistency. Please add a note regarding the long-term reliability of the lysimeter plots and if any re-calibration or soil refilling occurred.
Comments on the Quality of English Language

The manuscript is generally understandable and the technical content is communicated effectively. However, the English language should be improved to enhance clarity and academic tone. Several sentences are awkwardly constructed or contain redundancies, especially in the introduction and discussion sections (e.g., “It is imaginable that…”). A professional English editing service or careful language revision is recommended to ensure smoother readability, especially for international readers.

Reviewer 2 Report

Comments and Suggestions for Authors

The manuscript by Liu et al. investigated the effect of irrigation on crop production and nitrogen loss in sloping land with shallow soils, the study on the interrelationship between irrigation and nitrogen loss is of great significance for improving irrigation efficiency, promoting plant growth in shallow soil areas of sloping land, and increasing yield. However, the current version of the manuscript has some serious drawbacks need to be addressed.

1, The title of the manuscript mentioned crop production, however, the study was conducted just on the three types of crops, they can not represent all the crops, the title should be expressed more precisely. Besides, it seemed to be a simulated sloping land but not the field experiment.

2, The Abstract section does not clearly clarify the main objectives of this study and needs to be revised and improved. In addition, this section is limited to the description of results. What is the impact mechanism reflected by these results?
The Abstract need to be improved.

3, The introduction section needs to be reorganized, as the logical relationship between paragraphs is not clear enough. The contradictory and unified relationship among irrigation, yield increase, water-nitrogen loss, and irrigation infrastructure investment has not been clarified.

4, Line 59-66, this paragraph does not elaborate on the interaction between water and nitrogen in the soil during irrigation and its impact on plant nitrogen absorption and utilization, which needs to be improved.

5, Line 102, what dose the acronym "HF" refer to?

6, The discussion section fails to clearly elaborate on mechanisms or causes such as irrigation enhancing crop nitrogen absorption, water-nitrogen interaction under rainfall and different irrigation treatments, and how appropriate irrigation promotes crop nitrogen uptake while reducing nitrogen loss. The discussion is insufficiently in-depth and more resembles a repetitive restatement of the results.

7, Line 328-329, what was the basis for determining the occurrence of drought?

8, The methodology section does not clarify why the authors set different irrigation methods and planted different crops at different times. These factors may all cause changes in soil water and nitrogen loss. In addition, all treatments were subjected to the same rainfall, while the SI and FI treatments received additional irrigation. It remains unclear whether the amount and frequency of irrigation were sufficient to alter the trend of soil water and nitrogen loss caused by rainfall itself. Factors such as crop types, irrigation methods, irrigation amount, irrigation frequency, rainfall amount and frequency, and water-nitrogen interaction all affect soil water and nitrogen loss. Therefore, based on this experimental design, it is difficult to determine whether changes in soil water and nitrogen loss are caused by irrigation, and it is also challenging to optimize the irrigation system accordingly.

9, Line 397-401, irrigation methods, irrigation amount and frequency all have an impact on water and nitrogen loss, and the ANOVA cannot fully reflect the influence of each factor.

10, Line 409-410, without data on plant nitrogen content, the author cannot prove that irrigation has increased the absorption of nitrogen by plants.